# Clinical and Genetic Characteristics of Hypophosphatasia in Chinese Adults

**DOI:** 10.3390/genes14040922

**Published:** 2023-04-16

**Authors:** Xiang Li, Na Ren, Ziyuan Wang, Ya Wang, Yunqiu Hu, Weiwei Hu, Jiemei Gu, Wei Hong, Zhenlin Zhang, Chun Wang

**Affiliations:** 1Shanghai Clinical Research Center of Bone Disease & Department of Osteoporosis and Bone Diseases, Shanghai Sixth People’s Hospital Affiliated to Shanghai Jiao Tong University School of Medicine, Shanghai 200233, China; ideal13alt@hotmail.com (X.L.); mrna1960@163.com (N.R.); wangziyuan-96@sjtu.edu.cn (Z.W.); 18930173590@163.com (Y.W.); huyunqiu926@sohu.com (Y.H.); huweiwei1980@163.com (W.H.); gujiemei81@163.com (J.G.); 2Department of Osteoporosis & Shanghai Key Laboratory of Clinical Geriatric Medicine, Huadong Hospital, Fudan University, Shanghai 200040, China; drivyh@126.com; 3Clinical Research Center, Shanghai Sixth People’s Hospital Affiliated to Shanghai Jiao Tong University School of Medicine, Shanghai 200233, China

**Keywords:** hypophosphatasia, *ALPL*, mutations, China, adult

## Abstract

Hypophosphatasia (HPP) is an inherited disease caused by *ALPL* mutation, resulting in decreased alkaline phosphatase (ALP) activity and damage to bone and tooth mineralization. The clinical symptoms of adult HPP are variable, making diagnosis challenging. This study aims to clarify the clinical and genetic characteristics of HPP in Chinese adults. There were 19 patients, including 1 with childhood-onset and 18 with adult-onset HPP. The median age was 62 (32–74) years and 16 female patients were involved. Common symptoms included musculoskeletal symptoms (12/19), dental problems (8/19), fractures (7/19), and fatigue (6/19). Nine patients (47.4%) were misdiagnosed with osteoporosis and six received anti-resorptive treatment. The average serum ALP level was 29.1 (14–53) U/L and 94.7% (18/19) of patients had ALP levels below 40 U/L. Genetic analysis found 14 *ALPL* mutations, including three novel mutations—c.511C>G (p.His171Ala), c.782C>A (p.Pro261Gln), and 1399A>G (p.Met467Val). The symptoms of two patients with compound heterozygous mutations were more severe than those with heterozygous mutations. Our study summarized the clinical characteristics of adult HPP patients in the Chinese population, expanded the spectrum of pathogenic mutations, and deepened clinicians’ understanding of this neglected disease.

## 1. Introduction

Hypophosphatasia (HPP, OMIM 146300, 241500, 241510) is a rare metabolic skeletal disorder caused by loss-of-function mutation in the *ALPL* gene, which encodes the tissue-nonspecific isoenzyme of alkaline phosphatase (TNSALP), leading to decreased activity of alkaline phosphatase (ALP) and accumulation of endogenous substrates, including inorganic pyrophosphate (PPi), pyridoxal 5′-phosphate (PLP), and phosphoethanolamine (PEA). The pathogenesis of HPP is attributed to the accumulation of substrates. PPi overload results in impaired bone mineralization by inhibiting hydroxyapatite formation and causes calcium pyrophosphate dihydrate deposition disease (CPPD) in joints [1,2]. Therefore, patients mainly suffer from skeletal and dental diseases. The clinical manifestations of typical cases include exfoliation of the deciduous teeth, rickets, and repeated fractures. Moreover, excessive PLP may lead to pyridoxine-dependent seizures and other neuropsychiatric symptoms.

The age of onset and disease severity are highly correlated. From mild to severe, HPP is primarily categorized as odonto, adult, childhood, infantile, and perinatal types [2]. Clinical symptomatology of early-onset HPP is always severe and typical. In contrast, symptoms of adult HPP are milder and heterogeneous, ranging from fragility fractures and decreased bone mineral density (BMD) to generalized weakness. Because of the significant heterogeneity of clinical manifestations, this disease has been largely ignored and even misdiagnosed [1]. Some patients are misdiagnosed with primary osteoporosis and some even receive anti-resorptive therapies. Anti-resorptive agents, such as bisphosphonates, cause further deterioration of bone mineralization and increase the risk of atypical femoral fractures (AFFs) in HPP patients because HPP itself might also be a risk factor [3,4,5]. Compared with early-onset HPP, adult HPP has received less attention, and its clinical and genetic characteristics still need further study.

The purpose of this study is to describe the genetic and phenotypic characteristics of Chinese adults with HPP, explore the correlation between genotypes and phenotypes, expand the disease mutation spectrum, and deepen the understanding of HPP in clinicians.

## 2. Materials and Methods

### 2.1. Subjects

This retrospective study was designed in accordance with the principles of the Declaration of Helsinki and approved by the Ethics Committee of the Shanghai Sixth People’s Hospital (Approval number: 2019-117).

All patients with serum ALP levels lower than 40 U/L who presented to the Department of Osteoporosis and Bone Disease of Shanghai Sixth People’s Hospital between 2019 and 2022 were screened. A further study was conducted in 26 patients with serum ALP levels below 40 U/L at least twice and excluded from chronic kidney disease, hypothyroidism, anemia, and malnutrition. Sanger sequencing of the *ALPL* gene was performed on them and their available family members. One hundred healthy volunteers were sequenced and served as controls.

Fifteen patients and four direct family members with *ALPL* mutations were included and investigated. Among all 19 patients, there were two families (FAM1 and FAM2) and thirteen patients without family members participating in this study (represented by PT 1–13). All subjects were from nonconsanguineous Han ethnicity families. Informed consent was obtained from all participants.

### 2.2. Biochemical Analysis

Peripheral blood samples were collected after an overnight fast and analyzed in the central laboratory of Shanghai Sixth People’s Hospital. Serum calcium, phosphorus, and ALP were measured using a Hitachi 7600 automatic biochemical analyzer (Hitachi, Ltd., Tokyo, Japan). Serum 25-hydroxyvitamin D (25OHD), intact parathyroid hormone (PTH), and bone turnover markers (BTMs), including β cross-linked carboxy-terminal telopeptide of type I collagen (β-CTX) and osteocalcin (OC), were measured with a Roche electrochemiluminescence system (E170; Roche Diagnostic GmbH, Mannheim, Germany).

### 2.3. Radiography and Bone Densitometry

According to the needs, some patients underwent X-ray bone measurement in the Department of Radiology of Shanghai Sixth People’s Hospital.

Dual X-ray absorptiometry (GE Lunar Corp., Madison, WI, USA) was adopted for BMD measurement. The BMD (g/cm^2^) of the lumbar spine (L1–4), left femoral neck, and total hip was measured by a trained technician in the Department of Osteoporosis and Bone Disease of Shanghai Sixth People’s Hospital.

### 2.4. Genetic Analysis of ALPL

Blood samples were collected from patients and ethnically matched volunteers after an overnight fast for DNA analysis. Following the manufacturer’s instructions, genomic DNA was extracted from white blood cells in 2 mL of peripheral blood using an extraction kit (Shanghai Laifeng Biotechnology Co., Shanghai, China). Sanger sequencing of the *ALPL* gene was performed on patients and some family members to detect whether there was an *ALPL* gene mutation. The mutations were identified by matching the online database and comparing the genetic analysis of healthy volunteers. All 12 exons and exon–intro boundaries were amplified by PCR, and the primers were designed according to the previous literature [6]. Sequencing was performed using the BigDye Terminator Cycle Sequencing Ready Reaction Kit version 3.1 (Applied Biosystems, Foster City, CA, USA). The sequencing products were further analyzed by a sequencer (3130XL, Applied Biosystems, Foster City, CA, USA). The sequencing files were processed by PolyPhred software (https://droog.gs.washington. edu/polyphred/ (accessed on 16 January 2023)).

Three online tools, SIFT (https://sift.bii.a-star.edu.sg/ (accessed on 15 February 2023)), MutPred2 (http://mutpred.mutdb.org/ (accessed on 15 February 2023)), and MutationTaseter (https://www.mutationtaster.org/ (accessed on 15 February 2023)), were adopted for predicting the pathogenicity of novel mutations. AlphaFold2 was used to construct the 3D structure of mutated TNSALP [7]. Swiss-model (https://swissmodel.expasy.org/ (accessed on 28 March 2023)) was adopted to construct protein structures by homology modeling. Homology models were based on the crystal structure of human placental alkaline phosphatase (PDB number: 3mk1.1.A). Protein visualization, alignment, and intramolecular interaction predictions were performed by PyMol 2.5 (Schrödinger, New York, NY, USA).

### 2.5. Statistics

Student’s *t*-test was adopted to compare demographic parameters and laboratory findings between monoallelic heterozygous adult-onset HPP patients with and without a history of fracture. Multiple linear regression was used to determine the correlation between serum ALP and other laboratory parameters. Statistical analysis was performed by SPSS 22.0 (IBM, New York, NY, USA).

## 3. Results

### 3.1. Clinical Features of Subjects

The current study involved 2 families and 13 sporadic cases, for a total of 19 HPP patients. All patients were of Chinese Han ethnicity. The median age of the subjects was 62 (from 32 to 74) years. The ratio of males to females was 3 to 16. Except for the proband of family 1 (FAM-1-3), who was diagnosed with the childhood type, the other subjects were identified as adult-onset HPP. The clinical characteristics of all patients are summarized in Table 1.

Musculoskeletal symptoms were the most common clinical manifestations; more than half of our patients complained about muscular and skeletal symptoms (12/19, 63.2%). Six patients suffered from persistent backache and three patients endured the malaise of arthralgia; two patients reported leg pain, which turned out to be femoral pseudofractures (FAM1-3 and PT-13). One patient had an intermittent muscle spasm in her lower extremities (PT-1). FAM2-3, FAM2-4, PT-3, PT-5, and PT-12 had musculoskeletal symptoms ranging from years to decades.

Fragility fractures were also common in the current cohort. A total of ten incidents of nontraumatic fractures were identified, a great number of which were vertebral fractures (4/10, 40%). Family 1 proband (FAM1-3) experienced multiple nontraumatic fractures, including a Colles’ fracture of the right forearm, multiple vertebral fractures, and a pseudofracture on the left femur (Figure 1F). These fractures were confirmed by further X-ray bone survey. Multiple vertebral fractures were also identified in FAM2-4, the proband of the second family, and PT-10. PT-11 reported a history of vertebral compression fracture. PT-3 fractured her ankle at approximately 50 years old. PT-8 had a metatarsal stress fracture at approximately 40 and healed after plaster fixation. PT-13 initially presented to the Department of Orthopedics for pain in his right thigh. Radiological studies revealed a fracture of the right femur, and surgery was performed. However, his pain persisted and even extended to the other side after surgery. Further X-ray examination showed a pseudofracture on the left femur, hyperostosis of the pelvis and femurs, and CPPD (Figure 1A,B,H). CPPD was also identified in PT-12 (Figure 1C,H).

Dental issues were identified in nearly half of our patients (8/19, 42.1%). Only FAM1–3 reported the early loss of deciduous teeth and abnormal tooth eruption. FAM2-4 (Figure 1D,E), FAM2-5, PT-3, PT-5, PT-10, PT-12, and PT 13 provided a history of losing permanent teeth at age 40 or so. Dental caries and periodontitis were also identified in individuals.

Fatigue was frequently complained about by patients. Six patients (31.6%), most in their 30s or 40s, reported constant symptoms of fatigue at work or at rest. According to physical examinations, the proband of family 1 (FAM1-3) was characterized by waddling gait, genu valgum, and scoliosis. His height was 132 cm and his weight was 38 kg, with average intelligence and secondary sexual characteristics. Short stature was also identified in PT-1 (147 cm, age- and gender-specific reference average height: 158 cm), PT-10 (142 cm, age- and gender-specific reference average height: 155.7 cm), and PT-13 (153 cm, age- and gender-specific reference average height: 166.4 cm) [8]. 

Underdiagnosis and misdiagnosis were common in the current cohort. FAM1-3 received a confirmative diagnosis at the age of 38, with disease onset in childhood. FAM2-5 was potentially misdiagnosed as osteoarthritis. Misdiagnosis may also occur in PT-3 and PT-12, who were diagnosed with rheumatic arthritis before visiting our outpatient clinic. Moreover, nine patients were previously misdiagnosed with primary osteoporosis, leading to an astonishing misdiagnosis rate of 47.3%; six misdiagnosed patients were even treated with anti-resorptive agents. PT-8 and PT-10 received weekly alendronate for three and four years, respectively. PT-4 is a 43-year-old woman who was treated with yearly zoledronic acid for two years. PT-9, PT-11, and PT-12 were previously treated with denosumab.

### 3.2. Biochemical Parameters and BMD Measurement

The biochemical parameters of 19 HPP patients are summarized in Table 2. In this HPP cohort, the average serum ALP level (the most recent) was 29.1 (14–53) U/L. Except one subject (FAM1-2, the mother of the proband), 18 patients had serum ALP levels lower than 40 U/L. PT-8 was the only patient whose ALP level (14 U/L) was below the lower limit of normal set by our clinical laboratory (15 U/L). No patients had hypocalcemia or hypercalcemia, while only two patients (PT4 and PT5) had mild hyperphosphatemia. Vitamin D insufficiency and deficiency were common among our patients. The average serum level of 25OHD was 23.51 (11.9–31.1) ng/mL. Six patients were diagnosed with vitamin D deficiency because the serum level of 25OHD was lower than 20 ng/mL. Only one case of secondary hyperparathyroidism was found (FAM2-4). The levels of bone turnover varied from patient to patient; the levels of β-CTX and OC were 327.3 ± 200.34 ng/mL and 13.9 ± 4.55 ng/mL, respectively, because some of the patients had just received anti-resorptive therapies and FAM1-3 and PT-13 were in the acute phase of femoral pseudofractures.

According to the comparisons between monoallelic heterozygous patients (excluding two compound heterozygous subjects: FAM1-3 and PT-13) with and without a history of fracture (Table 3), the age of patients with fractures was significantly higher than that of patients without fractures (66.8 ± 3.56 vs. 50.75 ± 15.06, *p* = 0.004). The levels of ALP were lower in patients with a fracture history, but statistical significance was not identified (23.4 ± 6.47 vs. 31.50 ± 8.59, *p* = 0.059). Notably, three of the six patients with a fracture history received anti-resorptive therapies (PT-8, PT-10, and PT-11). A multiple linear regression failed to find a linear relationship between PTH, OC, β-CTX, 25OHD, and ALP levels after adjusting for the parameter age (sig = 0.646).

The BMD measurements of nine patients, including two premenopausal women and two men, are presented in Table 4. In this study, compared with the sex- and age-matched database, the BMD value of our adult HPP patients was only slightly decreased. The average Z scores of the lumbar spine, femoral neck, and hip were −1.27 (0.4 to −2.5), −1.26 (−0.5 to −1.8), and −1.2 (−0.4 to −1.8), respectively. The BMD of four postmenopausal women met the diagnostic criteria of osteoporosis (FAM2-4, FAM2-5, PT-3, and PT-9).

### 3.3. Mutation Analysis of ALPL

Altogether, 14 mutations were identified in this study (Table 5). The mutation analysis results of sporadic patients and HPP pedigrees are shown in Figure 2, and the mutation spectrum is shown in Figure 3. Eleven mutations were previously reported and included by either ClinVar (www.ncbi.nlm.nih.gov/clinvar/ (accessed on 15 February 2023)) or LOVD v3.0 (https://databases.lovd.nl/shared/genes/ALPL (accessed on 15 February 2023)). c.511C>G (p.His171Ala), c.782C>A (p.Pro261Gln), and c.1399A>G (p.Met467Val) were first described in the current study. c.1208G>A (p.Ser403Ala) was previously reported, but its clinical significance remains elusive. After the analysis of three major pathogenicity prediction tools, c.511C>G (p.His171Ala), c.1208G>A (p.Ser403Ala), and c.1399A>G (p.Met467Val) were predicted as disease-causing mutations, while c.782C>A (p.Pro261Gln) was predicted to have uncertain significance. Considering the typical symptoms of PT-3 (harboring the c.782C>A mutation), 3D models were constructed through in silico analysis of AlphaFold2 and Swiss-model to determine whether there was a structural alteration (Figure 4 and Figure 5). Intramolecular contact analysis was performed to predict the molecular effects of the *ALPL* mutations. It is predicted that replacing Pro261 with Gln261 residues will generate new hydrogen bonds between Gln261 and Tyr263 (Figure 4B,C). 

## 4. Discussion

HPP is an inherited skeletal disorder that results from loss-of-function mutation in the *ALPL* gene, which is located on chromosome bands 1p36.1–p34 and encodes TNSALP, the 50 kDa homodimer phosphohydrolase expressed predominantly in bone, liver, kidney, and teeth [2,21]. Since the first report of *ALPL* mutation in 1988, more than 500 potential pathogenic variants have been documented, with a predominance of missense mutations [2,21,22]. HPP is inherited as both an autosomal recessive and dominant trait. The estimated prevalence of severe recessive HPP in Canada and Europe is 1:100,000 and 1:300,000, respectively [23,24]. However, the prevalence of heterozygous HPP with mild symptoms is estimated to be as high as 1:508 in the European population [25,26].

Although the prevalence of HPP in China is still unknown, research on HPP is also receiving increasing attention. Xu, et al. [27] reported ten HPP patients and explored the dominant negative effect of identified missense mutations. Zhang, et al. [28] reported the clinical and genetic results of pediatric-onset HPP patients from five unrelated families and investigated the pathogenicity of mutations with in-depth bioinformatic analysis. Liu, et al. [29] reported the clinical, laboratory, and genetic findings of 33 pediatric-onset HPP patients and explored the differences between perinatal, infantile, childhood, and odonto subtypes. However, the characteristics and current situation of adult HPP in the Chinese population remain to be elucidated. The current study presented and discussed the clinical manifestations, laboratory findings, BMD results, genetic analysis, and misdiagnosis of 19 adult patients with HPP.

The severity of HPP varies among patients, ranging from prenatal death cases to almost asymptomatic cases. According to the data of the global HPP Registry, the common symptoms of adult HPP patients are chronic bone pain (52.5%), tooth problems (42.6%), fatigue (23.4%), and fractures (22%) [30]. According to a cohort study of German patients, headache (55%) is also a common complaint [10]. In addition, their physical function, social function, and mental health are also affected to some extent [30,31]. In the current study, 19 Han Chinese adults were included, of which 18 were adult HPP cases and 1 was a childhood HPP case. Musculoskeletal symptoms, such as arthralgia and backache, were the most common symptoms in these patients (63.2%). In addition to the early loss of permanent teeth (36.8%), there were frequent dental caries and periodontitis. Exacerbating pain in the leg is a clinical manifestation that requires extra attention because FAM1-3 and PT-13 both complained about leg pain, and the final radiographic diagnosis was femoral pseudofracture. The symptoms of our childhood HPP patient (FAM1-3) were typical. Growth retardation and early loss of deciduous teeth appeared in his childhood. With age, his leg gradually bent and developed a femoral pseudofracture. Although the clinical symptoms of adult HPP are milder than those of early-onset HPP, the burden of the disease is still significant.

According to the gene analysis results, 14 mutations were found in our patient cohort, including 11 previously reported mutations and 3 novel mutations. Bioinformatic analysis was used to analyze the pathogenicity of three novel mutations. c.511C>G (p.His171Ala) and c.1399A>G (p.Met467Val) were predicted to have pathogenicity. However, the clinical significance of c.782C>A (p.Pro261Gln) was still inconclusive because MutationTaster predicted it to be “polymorphic”, MutPred2 scored 0.351, and SIFT predicted significance was 0.32. Therefore, we further established the 3D model of the mutation by AlphaFold2 and Swiss-model to determine whether the alternation of a single amino acid will lead to structural changes. Significant alterations were not observed in models constructed by both in silico prediction tools. Based on a previous study, proline261 is a highly conserved residue located in the calcium-binding region [32]. Therefore, a slight change might lead to significant functional impairment of TNSALP. Proline is classified as a charge-neutral, nonpolar amino acid, while glutamine is a charge-neutral, polar amino acid. This polarity change may alter its interaction with adjacent residues. According to intramolecular contact analysis, new interactions were predicted to arise after the single amino acid alternation (Figure 4B,C), potentially causing impaired function of TNSALP. The clinical manifestations of PT-3 were typical, but the results of bioinformatic analysis were less significant. Therefore, further functional tests at the cellular level are necessary to clarify the pathogenicity of this specific point mutation. In the current study, c.407G>A (p.Arg136His) and c.984_ 986delCTT (p.Phe328del) were found in three and two unrelated patients, respectively, indicating the high prevalence of HPP in Chinese adults. c.407G>A (p.Arg136His) was also reported in other Chinese HPP cohorts [27,28,29].

Because of the broad spectrum of *ALPL* mutations and the high variability of phenotypes [1], it is challenging to establish a clear genotype–phenotype correlation. Consistent with previous experience, the clinical symptoms of patients with early onset and patients with biallelic mutations are more severe [26,33]. In this study, two carriers of compound heterozygous mutations (FAM1-3 and PT-13) showed more severe phenotypes. They both had femoral pseudofractures and dental problems. The heights of FAM-1-3 and PT-13 were 132 cm (age- and gender-specific reference average height: 169.7 cm) and 153 cm (age- and gender-specific reference average height: 166.4 cm), respectively [8]. Monoallelic mutations are associated with mild clinical symptoms. In this study, PT-2 and PT-9 with c.407G>A (p.Arg136His) heterozygous mutation were characterized by fatigue and decreased BMD. FAM1-2 and FAM1-4, who harbored monoallelic c.406C>T (p.Arg136Cys) mutations, had almost no clinical symptoms. Both c.406C>T and c.407G>A mutations lead to arginine replacement, so they may have similar biological effects. According to the report of Fauvert, et al. [34], the activity of p.Arg136His was 33.5% of that of wild-type TNSALP and should be regarded as a mild mutation. Combined with the literature and the results of this study, we speculate that adult patients harboring the heterozygous mutation c.406C>T or c.407G>A may be related to mild clinical symptoms.

Is there any abnormality in the serum levels of ALP and BTMs in adult patients? In the Danish cohort of 40 adult HPP patients, the serum levels of ALP, bone-specific alkaline phosphatase, and PLP were significantly decreased, but the levels of PTH, OC, and β-CTX were comparable to those in the age-matched control cohort [35]. The concentration of N-terminal pro-peptide of type 1 procollagen in HPP patients decreased significantly (44.2 ± 1.4 vs. 53.7 ± 1.4 µg/L, *p* = 0.006), indicating that bone turnover was inhibited; the phosphate level of patients in this cohort was higher than that in the control group, but it was still within the normal reference range (1.16 (1.04–1.29) vs. 1.01 (0.92–1.15) mmol/L, *p* = 0.002), and the calcium level between the groups was similar [35]. In our study, two patients (PT-4 and PT5) had mild hyperphosphatemia, indicating impaired bone mineralization. The difference in BTMs between patients was not discussed in the current study because six patients received anti-resorptive treatment and two patients were in the fracture acute phase. According to the lower limit of normal provided by our clinical lab (15 U/L), only one patient’s (PT-8) ALP level was regarded as decreased (14 U/L). According to proposals by other scholars [4,36], 40 U/L might be an ideal threshold. Using this threshold, eighteen of our patients were diagnosed with decreased serum ALP. Decreased ALP is the hallmark of HPP and is critical for diagnosis. Therefore, it is vital to establish a reference range according to age and gender to avoid misdiagnosis and underdiagnosis. According to the comparison between monoallelic mutation patients, ALP levels appeared to be associated with the risk of fracture. However, the correlation disappeared after age correction. In adult HPP, the occurrence of fractures and multiple symptoms is associated with increased concentrations of the ALP substrate PLP, while there are no significant differences in ALP levels between patients with and without fracture as well as patients with fewer symptoms and multiple symptoms (more than two symptoms) [10]. 

According to a literature review, the BMD of adults with HPP only slightly decreased [37,38,39]. In contrast, higher vertebral BMD (possibly owing to excessive PPi deposition) is associated with an increased risk of fractures in adult HPP [37]. In this study, the BMD values of adult HPP patients were lower than those of the age- and sex-matched database, but a correlation between lumbar BMD and disease severity was not found.

Because of the high variability in clinical symptoms, it is difficult to diagnose adult HPP at present. According to the experience of a single rheumatology center, 46.1% of HPP patients have no obvious clinical manifestations [40]. The problem of underdiagnosis and misdiagnosis of adult HPP is increasingly prominent. According to the global HPP Registry, the average age at diagnosis of adult HPP is 48.9 years old [30]. A study showed that the average interval from symptoms to diagnosis was 14.4 years [39]. On the other hand, the prevalence of HPP is high. In osteoporosis clinics, the prevalence of HPP was as high as 0.3–0.43% [41,42]. Misdiagnosis is common in adult HPP [33,42,43,44]. In our study, two patients were potentially misdiagnosed with RA and nine patients were misdiagnosed with primary osteoporosis. Among these nine patients, six received anti-resorptive treatment, including bisphosphonates and denosumab. One patient (PT-10) took weekly alendronate sodium for up to 4 years. AFF is a symbol of HPP [3]. However, if HPP patients are misdiagnosed with osteoporosis and receive anti-resorptive treatment, they will face greater risks of AFF. Lefever et al. [33] reported that two patients with HPP developed AFF after receiving anti-resorptive therapy. Zhou, et al. [45] reviewed the occurrence of AFF in monogenic bone diseases and found 23 HPP patients with AFF, of whom 10 had been exposed to bisphosphonates. In another single-center, cross-sectional study involving 150 HPP patients, 46.7% of AFF patients were exposed to bisphosphate for an average duration of 5.33 years (1–11 years) [4]. Fortunately, none of our patients who received anti-resorptive therapy developed AFF.

Typical clinical manifestations of both adult HPP and primary osteoporosis include fragility fracture and low BMD. Therefore, when a patient experiences the following conditions, HPP should be suspected, and further evaluations should be considered to avoid misdiagnosis and underdiagnosis: (1) the serum level of ALP remained consistently lower than 40 U/L; (2) multiple fragility fractures; (3) AFF with and without previous anti-resorptive treatment; and (4) dental issues: early loss of permanent teeth, sparse teeth, and so on.

Adult-onset HPP lacks pharmacological treatment. Enzyme replacement therapy is only approved for adults with pediatric-onset HPP, excluding adult HPP patients. However, if fractures in adult patients do not heal, enzyme replacement therapy may be considered. Magdaleno, et al. reported a debilitated woman with adult-onset HPP whose symptoms of pain and motor function improved significantly after 10 months of treatment with asfotase alfa [46]. Some studies have also shown that asfotase alfa may improve fracture healing in adults with childhood-onset HPP [47,48].

*ALPL* is expressed by osteoblasts [49]. Therefore, the anabolic agent teriparatide may be a good choice for adult HPP. In 2007, Whyte, et al. [50] reported the first case successfully treated with teriparatide. A 56-year-old white woman suffered from multiple metatarsal stress fractures and proximal femoral fractures. After six weeks of treatment, her pain was relieved. Her multiple fractures gradually healed. ALP and BTMs continued to increase during 18 months of teriparatide treatment [50]. Another study showed that the healing-promoting effect still existed when the patient encountered a new fracture and initiated another episode of therapy [51]. At the histological level, 6 months of treatment with teriparatide has been proven to almost reverse osteomalacia and increase bone formation [52]. However, other studies reported the opposite results. Laroche [53] reported a female patient whose symptoms worsened after 10 months of treatment. Although the serum level of β-CTX increased threefold, her serum level of ALP remained low. A lack of sustained response to teriparatide was found in another adult HPP [54]. In our study, PT-11 was treated with teriparatide for 3 months. Her serum ALP and OC levels were significantly increased from 19.2 to 48 U/L and 8.6 to 51.66 ng/mL, respectively. Her musculoskeletal symptoms were also improved. Considering the enormous heterogeneity of genetic variation in HPP patients, it is speculated that the efficacy of teriparatide may be related to different pathogenic mutations. The benefits, risks, and duration of teriparatide treatment need to be further clarified through large-sample-size studies with long-term follow-up.

The current study has the following limitations. First, the sample size of the study is limited, which is not sufficient to determine the genotype–phenotype correlation of Chinese adult HPP. Second, TNSALP substrates, namely PLP and PEA, were not measured owing to the limited blood samples.

## 5. Conclusions

This study presented and discussed the clinical and genetic characteristics of HPP in Chinese adults and identified three new *ALPL* mutations. Two patients carried biallelic mutations, showing a more severe phenotype than monoallelic mutations. Nearly half of HPP patients were misdiagnosed with osteoporosis and some patients were even mistakenly treated with anti-resorptive agents. At present, adult HPP lacks drug treatment. One patient in this study received teriparatide. After three months of treatment, the level of ALP increased and the patient’s symptoms improved. Therefore, it may be an effective treatment option for some HPP patients.

## Figures and Tables

**Figure 1 genes-14-00922-f001:**
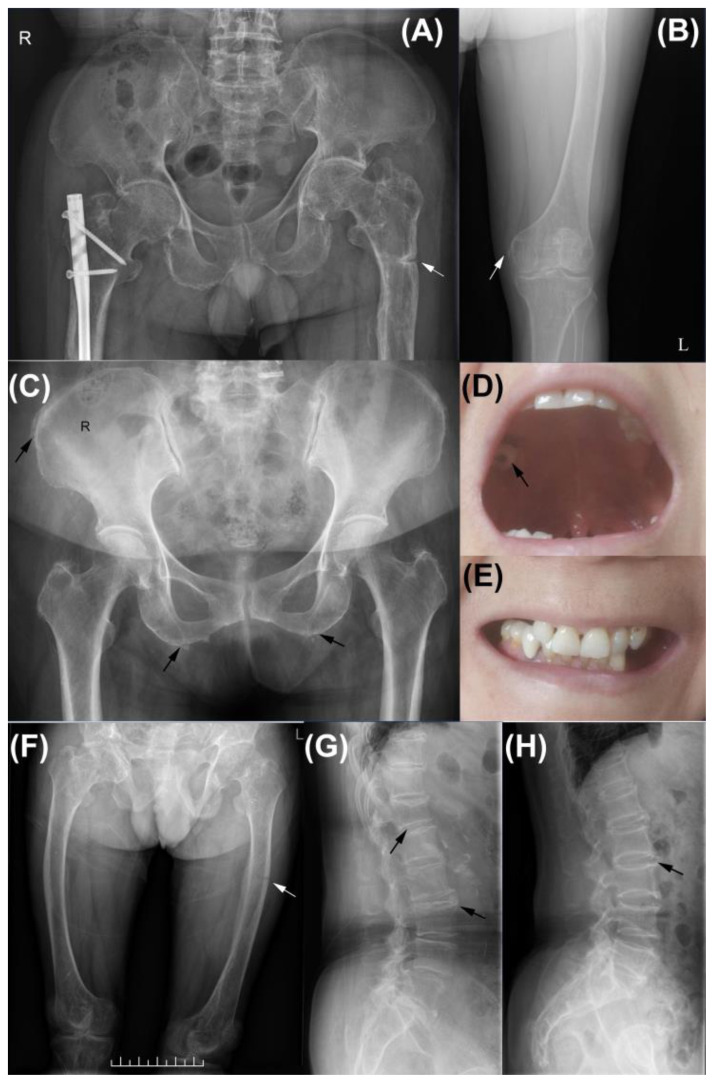
Radiological and dental findings of patients. (**A**) PT-13 previously experienced an atraumatic femoral fracture and received internal fixation, and a pseudofracture was identified on his left femur (white arrow); (**B**) calcium pyrophosphate dihydrate deposition disease (CPPD) of the left knee in PT-13; (**C**) pelvic CPPD found in PT-12, calcium pyrophosphate deposition was identified (black arrow); (**D**) sparse teeth, a dental decay, and (**E**) hypocalcified enamel were prominent in FAM2-3; (**F**) bowling of bilateral femurs and a pseudofracure (white arrow) were identified in FAM1-3; vertebral CPPD was identified in PT-12 (**G**) and PT-13 (**H**).

**Figure 2 genes-14-00922-f002:**
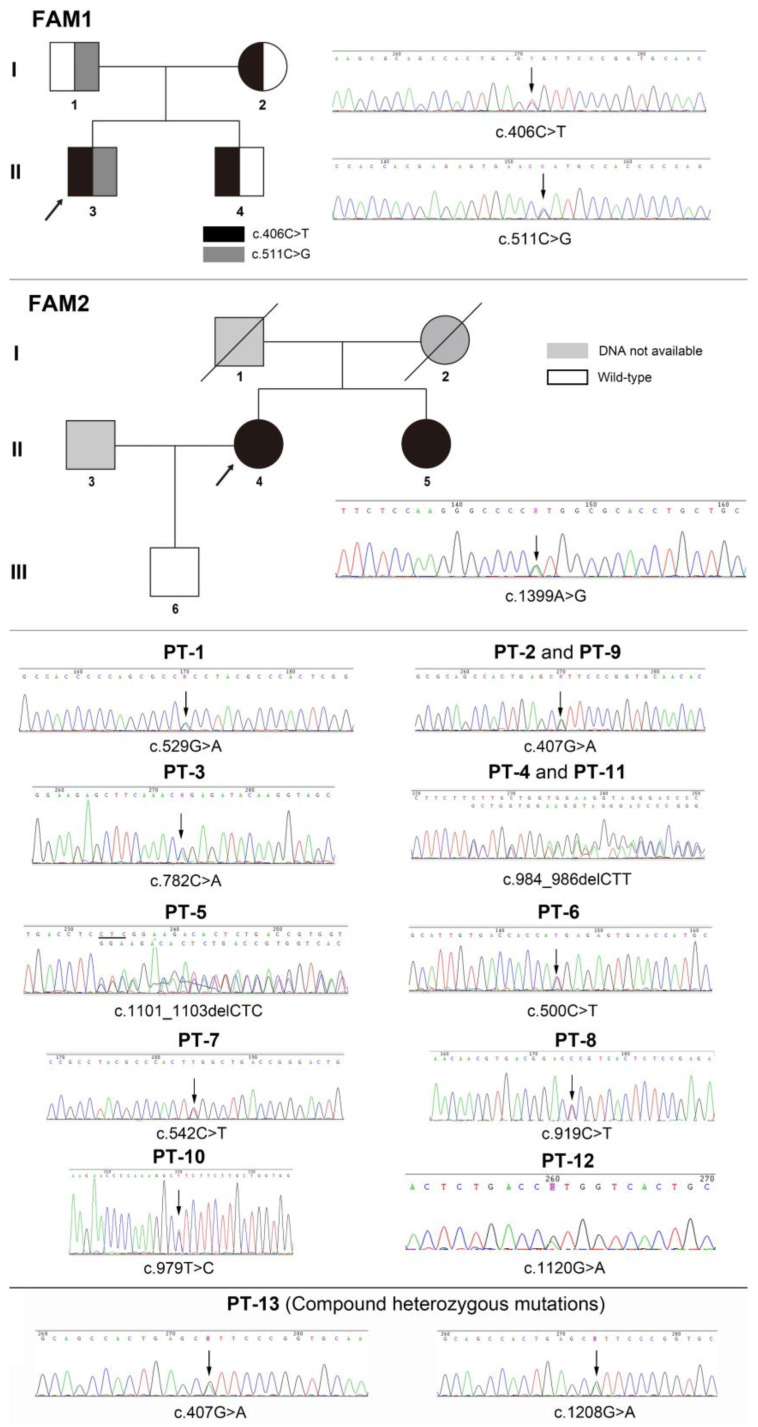
Genetic analysis of *ALPL* in familial and sporadic patients.

**Figure 3 genes-14-00922-f003:**
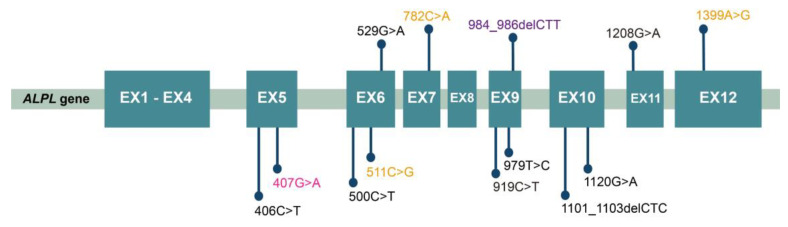
The mutation spectrum of *ALPL* in all patients. Mutations repeated three times are shown in rose red. Mutations repeated twice are shown in purple. Yellow font indicates the novel mutations first reported in the current study. Note:Intrafamilial repetition of the same mutation was regarded as appearing once.

**Figure 4 genes-14-00922-f004:**
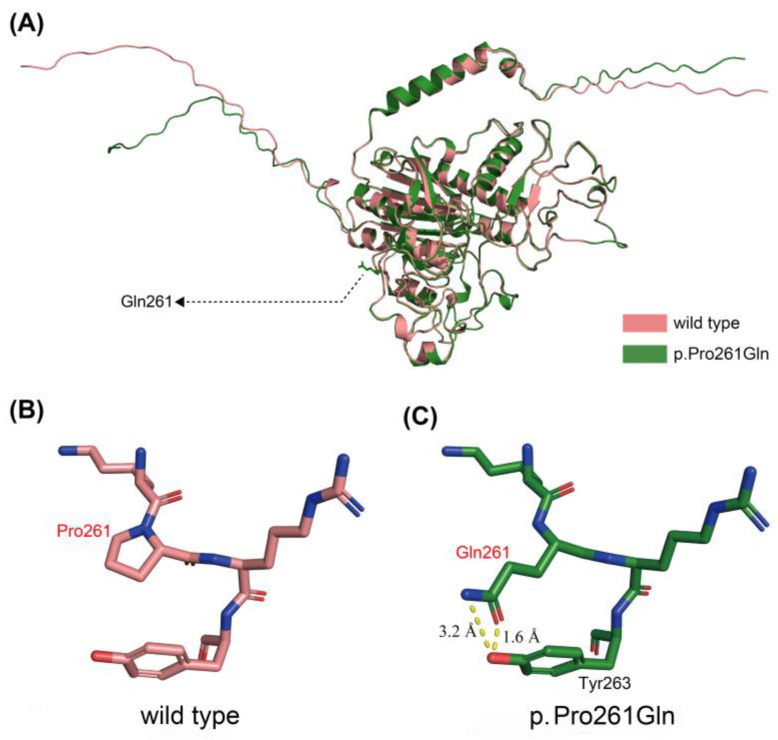
Three-dimensional structures of wild-type and Pro261Gln TNSALP constructed by AlphaFold2. (**A**) Alignment of wild-type TNSALP (downloaded from https://alphafold.ebi.ac.uk/entry/P05186 (accessed on 18 February 2023)) and p.Pro261Gln mutant. (**B**) Intramolecular analysis of wild-type TNSALP. (**C**) Intramolecular analysis revealed new interactions between Gln261 and Tyr263 (yellow dotted lines).

**Figure 5 genes-14-00922-f005:**
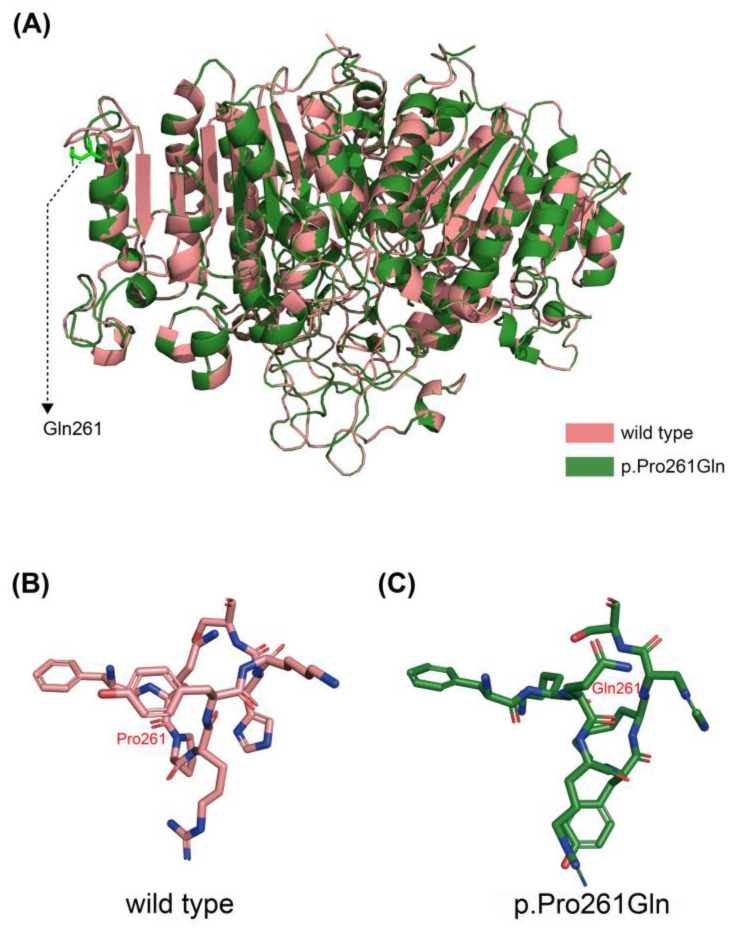
Three-dimensional structures of wild-type and Pro261Gln TNSALP constructed by Swiss-model. (**A**) Alignment of wild-type TNSALP and p.Pro261Gln mutant. (**B**,**C**) Intramolecular analysis of wild-type and mutant TNSALP, with no new interactions identified.

**Table 1 genes-14-00922-t001:** Clinical manifestations.

Patient	Sex	Age	Dental Issues	Fracture	Musculoskeletal Symptoms	Other	Misdiagnosis
FAM1-3	male	38	early loss of deciduous teeth, abnormal teeth eruption	RF, mVF, pseudofracture	leg pain	growth retardation, bowing legs	-
FAM1-1	male	64	-	-	-	fatigue	-
FAM1-2	female	64	-	-	-	-	-
FAM1-4	male	36	-	-	-	-	-
FAM2-4	female	70	early loss of permanent teeth, dental caries	mVF	backache		-
FAM2-5	female	63	early loss of permanent teeth, periodontitis	-	arthralgia	-	OA *
PT-1	female	34	-	-	muscle spasm	fatigue, short stature	
PT-2	female	32	-	-	-	fatigue	OP
PT-3	female	62	dental caries, early loss of permanent teeth	AF	arthralgia	fatigue	RA *
PT-4	female	43	-	-	-	-	OP ^Δ^
PT-5	female	58	early loss of permanent teeth	-	backache	-	OP
PT-6	female	32	-	-	-	fatigue	-
PT-7	female	62	-	-	arthralgia	-	OP
PT-8	female	64	-	MF	backache	-	OP ^Δ^
PT-9	female	47	-	-		fatigue	OP ^Δ^
PT-10	female	69	early loss of permanent teeth	mVF	backache	short stature	OP ^Δ^
PT-11	female	69	-	VF	backache	-	OP ^Δ^
PT-12	female	74	early loss of permanent teeth	-	backache	CPPD	OP ^Δ^, RA *
PT-13	female	54	early loss of permanent teeth	FF, pseudofracture	leg pain	CPPD, short stature	-

RF, radius fracture; mVF, multiple vertebral fractures; AF, ankle fracture; MF, metatarsal fracture; VF, vertebral fracture. OP, osteoporosis. OA *, suspected misdiagnosis of osteoarthritis. RA *, suspected misdiagnosis of rheumatic arthritis; ^Δ^ treated with anti-resorptive agent.

**Table 2 genes-14-00922-t002:** Laboratory findings.

Patients	ALP (U/L)			Calcium * (mmol/L)	Phosphate ^+^ (mmol/L)	PTH (pg/mL)	25OHD (ng/mL)	β-CTX (ng/mL)	OC (ng/mL)
	Recent	Average	Lowest						
FAM1-3	34	30.7	28	2.34	1.43	30.37	27.05	285.5	*13.33*
FAM1-1	40	NA	NA	2.3	1.31	38.37	22.35	328.5	NA
FAM1-2	53	NA	NA	2.23	1.51	39.34	*15.94*	286.4	NA
FAM1-4	32	NA	NA	2.33	1.25	34.2	*12.97*	84.96	NA
FAM2-3	28	29.3	26	2.32	1.21	48.32	26.79	267	15.66
FAM2-4	38	33.5	29	2.27	1.33	**65.43**	25.93	200.8	*14.44*
PT-1	23	28	22	2.3	0.94	49.2	*11.9*	237.7	*11.1*
PT-2	23	28	23	2.27	1.35	48.1	*18.88*	300.85	*13.67*
PT-3	27.8	27.6	25	2.4	1.08	20.5	25	657	15.7
PT-4	27	29	25	2.28	**1.64**	28.7	31.1	478.1	18.6
PT-5	32	30	28	2.22	**1.61**	29.3	*19.9*	NA	NA
PT-6	26	31.3	26	2.3	1.13	18.56	25.7	345	*12.8*
PT-7	29	28	27	NA	NA	NA	NA	NA	NA
PT-8	*14*	17	14	2.28	1.21	37.56	28.17	80.6	*9.27*
PT-9	27	31	27	2.49	1.12	43.41	26.52	274.9	*9.19*
PT-10	28	27.3	22	2.32	1.29	28.2	30.6	532	26.1
PT-11	19.2	20.4	19.2	2.3	1.12	55.9	*18*	100	*8.6*
PT-12	28	27.7	23	2.33	1.15	57.02	28.58	267.6	*11.05*
PT-13	23	24.3	22	2.36	1.33	18.64	27.8	836.4	15.51

ALP, alkaline phosphatase (reference provided by our clinical lab: 15–112 U/L); PTH, parathyroid hormone (reference: 15–65 pg/mL); 25OHD, 25-hydroxyvitamin D (reference: >20 ng/mL); β-CTX, β cross-linked carboxy-terminal telopeptide of type I collagen (reference: ˂1008 ng/L); OC, osteocalcin (reference: 15–46 ng/mL); NA, not available. Bold font indicates that the value is higher than the reference range; italic font indicates that the value is lower than the reference range. * reference range: 2.08–2.60 mmol/L; ^+^, 0.80–1.60 mmol/L.

**Table 3 genes-14-00922-t003:** Laboratory differences between monoallelic heterozygous patients with and without fracture.

	With Fracture	Without Fracture	*p*
Age, years	66.8 ± 3.56	50.75 ± 15.06	0.004
Female/all sex	5/5	11/12	-
ALP, U/L	23.4 ± 6.47	31.50 ± 8.59	0.059
Calcium, mmol/L	2.32 ± 0.046	2.30 ± 0.069	0.416
Phosphorus, mmol/L	1.18 ± 0.046	1.30 ± 0.209	0.129
PTH, pg/mL	38.09 ± 14.40	40.77 ± 12.92	0.712
25OHD, ng/mL	25.71 ± 4.77	22.33 ± 6.37	0.257
β-CTX, ng/mL	327.32 ± 238.25	280.48 ± 101.42	0.713
OC, ng/mL	15.07 ± 7.03	13.11 ± 2.90	0.578

PTH, parathyroid hormone (reference: 15–65 pg/mL); 25OHD, 25-hydroxyvitamin D (reference: >20 ng/mL); β-CTX, β cross-linked carboxy-terminal telopeptide of type I collagen (reference: <1008 ng/L); OC, osteocalcin (reference: 15–46 ng/mL).

**Table 4 genes-14-00922-t004:** Results of bone mineral density in nine individuals.

Patients	Age	LS BMD	Z Score	FN BMD	Z Score	TH BMD	Z Score
FAM1-3	38	0.744	−1.8	UM	-	UM	-
FM2-4	70	0.901	NA	0.452	NA	0.67	NA
FM2-5	63	0.895	NA	0.544	NA	0.77	NA
PT-1	34	0.804	−1.7	0.611	−1.8	0.719	−1.8
PT-2	32	1.111	0	0.711	−1.8	0.785	−1.5
PT-3	62	0.83	0.4	0.529	−1.3	0.704	−0.2
PT-4	43	0.892	−1.0	0.751	−0.5	0.869	−0.4
PT-9	47	0.792	−2.5	0.764	−0.9	0.66	−2.1
PT-13	54	0.921	−1.1	UM	-	UM	-

LS, lumbar spine; BMD, bone mineral density (g/cm^2^); FN, femoral neck; TH, total hip; UM, unmeasurable. NA, not available.

**Table 5 genes-14-00922-t005:** Genetic analysis of the *ALPL* gene in 19 patients.

Patients	Status	DNA Variations	Amino Acid Change	Reported	ACMG [9] Classification
FM1-3 PB	cHet/MS	c.406C>T	p.Arg136Cys	Yes	Pathogenic [10]
		c.511C>G	p.His171Ala	No	LP ^a^
FM1-1 PB’s farther	Het/MS	c.511C>G	p.His171Ala	No	LP ^a^
FM1-2 PB’s mother	Het/MS	c.406C>T	p.Arg136Cys	Yes	Pathogenic [10]
FM1-4 PB’s brother	Het/MS	c.406C>T	p.Arg136Cys	Yes	Pathogenic [10]
FM2-3 PB	Het/MS	c.1399A>G	p.Met467Val	No	LP ^b^
FM2-4 PB’s s sister	Het/MS	c.1399A>G	p.Met467Val	No	LP ^b^
PT-1	Het/MS	c.529G>A	p.Ala177Thr	Yes	Pathogenic [11]
PT-2	Het/MS	c.407G>A	p.Arg136His	Yes	Pathogenic [12]
PT-3	Het/MS	c.782C>A	p.Pro261Gln	No	US ^c^
PT-4	Het/DEL	c.984_986delCTT	p.Phe328del	Yes	Pathogenic [13]
PT-5	Het/DEL	c.1101_1103delCTC	p.Ser368del	Yes	Pathogenic [14]
PT-6	Het/MS	c.500C>T	p.Thr167Met	Yes	Pathogenic [15]
PT-7	Het/MS	c.542C>T	p.Ser181Leu	Yes	Pathogenic [16]
PT-8	Het/MS	c.919C>T	p.Pro307Ser	Yes	Pathogenic [17]
PT-9	Het/MS	c.407G>A	p.Arg136His	Yes	Pathogenic [12]
PT-10	Het/MS	c.979T>C	p.Phe327Leu	Yes	LP [18]
PT-11	Het/DEL	c.984_986delCTT	p.Phe328del	Yes	Pathogenic [13]
PT-12	Het/MS	c.1120G>A	p.Val327Met	Yes	LP [19]
PT-13	cHet/MS	c.407G>A	p.Arg136His	Yes	Pathogenic [12]
		c.1208G>A	p.Ser403Ala	Yes	US [20]

PB, proband; cHet, compound heterozygous; MS, missense mutation; LP, likely pathogenic; Het, heterozygous; US, uncertain significance; DEL, deletion mutation. Italic font indicates novel mutations. ^a^ c.511C>T (p.His171Thr) mutation was reported to be pathogenic in LOVD, c.512A>C (p.His171Arg) was reported to be pathogenic in ClinVar. Evidence of ACMG classification: PM2 + PM5 + PP3 + PP2 + PP4. ^b^ c.782C>A (p.Pro261Gln) was previously described by dbSNP (rs765149569) and c.782C>T (p.Pro261Leu) was reported to have uncertain significance in ClinVar. Evidence of ACMG classification: BP6 + PP2 + PP4. ^c^ c.1400T>C (p.Met467Thr) was reported to be pathogenic by LOVD. Evidence of ACMG classification: PM2 + PM5 + PP3 + PP2 + PP4.

## Data Availability

The data presented in this study are available on reasonable request from the corresponding authors. The data are not publicly available due to privacy.

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
