# Peer review of "Clinical and Genetic Characteristics of Hypophosphatasia in Chinese Adults"

_genes, 2023, doi:10.3390/genes14040922_

Round 1

Reviewer 1 Report

The complete review can be found in the attached pdf file.

Author Response

Dear Professor,

We want to thank you for your valuable comments which helped us improve the manuscript. It was more than a review, but a guideline for us to improve the preciseness and quality of the current work. We have learned a great deal through these comments and will improve our work in future scientific research and clinical practice. Thank you sincerely!

Results

  1. The authors reported that except for PT-8 patient the ALP serum levels were within limits of normal (table 1). However, in the material and methods section, they reported that “All the patients with low serum ALP levels who presented to the Department of Osteoporosis and Bone Disease of Shanghai Sixth People's Hospital between 2019 to 2022 were screened. Twenty-six patients with persistently low serum ALP levels were found.”

1.1 Does this indicate that these patients previously presented low serum ALP levels? Could the treatment that some of them received affect the recent evaluation of serum ALP levels?

We apologize for the ambiguity in the Materials and Methods section. According to the ALP reference range provided by our clinical laboratory, the lower limit of normal (LLN) is 15 U/L. Based on our experience and the threshold recommended by Genest et al. and Quinn et al. in previous studies, we adopted 40 U/L as the threshold of decreased serum ALP in this study.

In this study, Sanger sequencing of the ALPL gene was performed in 26 patients with ALP levels below 40 U/L at least twice. Among them, 11 patients did not detect ALPL mutations, and most of them had a history of treatment with anti-resorptive drugs. Therefore, the reason for the decrease in ALP may be related to anti-resorptive treatment. The remaining 15 patients and 4 family members were confirmed to have mutations in the ALPL gene, so a total of 19 subjects were included in this study. Among them, 18 patients had ALP levels below 40U/L. Only the mother (FAM1-2) of proband FAM1-3, had an ALP level (53 U/L) above the threshold value (40 U/L).

[1] Genest F, Seefried L. Subtrochanteric and diaphyseal femoral fractures in hypophosphatasia-not atypical at all. Osteoporos Int. 2018 Aug;29(8):1815-1825.

[2] Quinn HB, Busch RS, Kane MP. The Occurrence and Burden of Hypophosphatasia in an Ambulatory Care Endocrinology Practice. Endocr Pract. 2021 Dec;27(12):1189-1192.

Under your guidance, we revised our wording and descriptions in the relevant sections:

(Page 2, line 63-67)

“All patients with serum ALP levels lower than 40 U/L who presented to the Department of Osteoporosis and Bone Disease of Shanghai Sixth People's Hospital between 2019 and 2022 were screened. A further study was conducted in 26 patients with serum ALP levels below 40U/L at least twice and excluded from chronic kidney disease, hypothyroidism, anemia, and malnutrition”

(Page 7, line 14-18)

“In this HPP cohort, the average serum ALP level (the most recent) was 29.1 (14 - 53) U/L. Except one subject (FAM1-2, the mother of the proband), 18 patients had serum ALP levels lower than 40 U/L. PT-8 was the only patient whose ALP level (14 U/L) was below the lower limit of normal set by our clinical laboratory (15 U/L)”

1.2 Since that, low serum levels are considered a hallmark of hypophosphatasia (Riancho-Zarrabeitia, 2016), these controversial findings should be better discussed.

We agree with your opinion. Likewise, decreased ALP is the hallmark of our patients. The current study adopted an ALP level lower than 40 U/L to screen potential HPP patients. Again, we apologize for the ambiguous statement we made in the manuscript. By adopting 40 U/L as the LLN, 94.7% (18/19) of patients met the criteria of decreased ALP. If according to the LLN (15U/L) provided by our clinical laboratory, only PT-8 can diagnose decreased ALP (14U/L). An inappropriate reference range of ALP potentially leads to misdiagnosis and underdiagnosis of HPP. Therefore, establishing age- and sex-specific reference ranges of ALP is vital and is one of the aims of our study.

We have added relevant content in the discussion section

(Page 13, line 182 - 187)

“According to the lower limit of normal provided by our clinical lab(15U/L), only one patient's (PT-8) ALP level was regarded as decreased (14U/L). According to proposals by other scholars[4,36], 40 U/L might be an ideal threshold. Using this threshold, eighteen of our patients were diagnosed with decreased serum ALP. Decreased ALP is the hallmark of HPP and is critical for diagnosis. Therefore, it is vital to establish a reference range according to age and gender to avoid misdiagnosis and underdiagnosis.”

1.3 Diagnosis based on ALP serum level results could be one of the reasons why many patients in the cohort were underdiagnosis and misdiagnosis? It would be important additionally evaluate the ALP substrates levels such as phosphoethanolamine [PEA], inorganic pyrophosphate [PPi], and pyridoxal 5'-phosphate [PLP], however, in the discussion the authors declare that “ALPL substrates, namely PLP and PEA, were not measured due to the limited blood samples”.

HPP, especially adult-onset HPP, is highly overlooked in clinical practice. Diagnosis solely based on reference value provided by clinical laboratories (not age- and gender-specific), insufficient inquiry of medical and family history, inaccessibility to measure ALP substrates, and insufficient understanding of HPP are all reasons for the underdiagnosis and misdiagnosis.

Indeed, the measurement of PEA and PLP is important to establish a confirmative diagnosis. Because the current study is a retrospective study, it is regrettable that serum samples cannot be obtained for ALP substrate determination. Based on your valuable comments, we plan to collect both blood and urine samples of patients in subsequent follow-ups to consolidate the diagnosis.

  1. In table 5, the authors described the c.782C>A mutation as “No” reported and “Unknown” pathogenicity, but the other two (c.511C>T and c.1399A>G) “No” reported mutations were considered “pathogenic” based on in silico prediction. Is not clear to me, why the pathogenicity for c.782C>A variant can’t be predicted by bioinformatic tools? Whether the predictions were done or the results are not consistent this should be declared. The author should consider substituting the “Unknown” word for a more suitable word.

    We apologize for the misnomer. The pathogenicity of c.782C>A was also predicted by in silico analysis, and the results showed unknown significance. We replaced “Unknown” with the ACMG terminology “uncertain significance” (Page 8, table 5, line 52).

  1. I have some comments about the mutations reported as new in the study.

Note 1: Although there is no description for c.511C>G mutation in ClinVar (www.ncbi.nlm.nih.gov/clinvar/) or LOVD v3.0 (https://databases.lovd.nl/shared/genes/ALPL) as stated by authors, It is important to highlight that there is one mutation in the same nucleotide position (c.511C>T), resulting in a different change in protein level (p.His171Try) was previously described in an infantile case (genotype c.[511C>T];[571G>A]). Furthermore, ALPL mutation resulting in His replacement to Arg residue at protein 171 position (p.H171A) was reported by other authors in adult-HPP (genotype: c.[512A>G];[=] and c.[512A>G];[571G>A]) and Infantile (genotype: c.[512A>G];[542C>T]) HPP cases. I think that this information should be provided to the reader.

Note 2: There is no description in ClinVar or SNP database for c.1399A>G mutation (p.M467V), but a mutation affecting the same amino acid residue (p.M467T) was detected in an infantile HPP case (genotype: c.[919C>T];[1400T>C]) according Versailles lab (Oct. 2004; https://alplmutationdatabase.jku.at/table/).

Note 3: The c.782C>A mutation (p.P261Q) was previously described in dbSNP (rs765149569), and ClinVar also contains an entry for this variant (Variation ID: 388169). Furthermore, there is also a description for a mutation affecting the same residue 261 (NP_079420.3:p.Pro261Leu, dbSNP: rs745934168).

We appreciate your valuable comments. The above important information is added as captions to Table 5.

(Page 9, line 55-59)

ac.511C>T (p. H171T) mutation was reported to be pathogenic in LOVD, c.512A>C (p. p. H171R) was reported to be pathogenic in ClinVar. bc.782C>A (p. P261Q) was previously described by dbSNP (rs765149569) and c.782C>T (p. P261L) was reported to have uncertain significance in ClinVar.cc.1400T>C (p. M467T) was reported pathogenic by LOVD.”

Based on these crucial details, we provided ACMG classification for all three novel mutations (Page 8, table 5, line 52).

  1. In topic 3.3, sentence “…c.511C>G (p.H171A), c.782C>A (p.P261Q), and 1399A>G (M467V)”, the variants nomenclature is not standardize. The author should standardize the citation in all text. I suggest that the authors adopt the HGVS sequence variant nomenclature (https://varnomen.hgvs.org/) or another defined by periodic.

    Thank you for the suggestion. We unified and standardized the description of mutations in the revised manuscript.

  1. Gene name (ALPL) in some parts of the text is not in italic. For example, Abstract, figure 3 title, page 2 - line63, page 14 - line 230.

    We apologize for our carelessness. The abovementioned mistakes were corrected in the revised manuscript.

  1. The authors mention the abbreviation TNSALP (Material and Methods – 2.4 Genetic analysis of ALPL, page 3 and figure 4) before they write the complete protein name, which appears only in the discussion “… tissue-non-specific isoenzyme of alkaline phosphatase (TNSALP)”

We apologize for the mistake. We corrected the error and provided the full name of TNSALP in the revised Introduction section:

(Page 1, line 32-33)

“Hypophosphatasia (HPP, OMIM 146300, 241500, 241510) is a rare metabolic skeletal disorder caused by loss-of-function mutation in the ALPL gene, which encodes tissue‑nonspecific isoenzyme of alkaline phosphatase (TNSALP), leading to decreased activity of alkaline phosphatase (ALP) and accumulation of endogenous substrates, including inorganic pyrophosphate (PPi), pyridoxal 5’-phosphate (PLP), and phosphoethanolamine (PEA)”

  1. I would like to know why the authors have chosen Alphfold2 to predict the 3D structure since the crystal structure of human placental alkaline phosphatase, which shows ~74% of homology with TNSALP has been already previously determined? What model would be more confident for the prediction of functional and structural impact, homology or Alphfold2 (artificial intelligence)?

Note: In the last 20 years, several authors have built TNSALP 3D models using programs for homology modeling. These models have been based on the previously determined crystal structure of human placental alkaline phosphatase, which shows ~74% of homology with TNSALP. Because Alphfold2 is a new tool to predict 3D models of protein structure and think that TNSALP novel mutation should be modeled by both tools/methods, and the models should be compared.

We chose Alphfold2 for two reasons: (1) the crystal structure of the tissue-nonspecific isoenzyme of alkaline phosphatase (TNSALP) has not yet been determined by X-ray crystallography; (2) the new methodology AlphaFold 2 shows great prospects in predicting protein structures.

To be honest, our knowledge of structural biology is limited. It is difficult to judge which model is better. Therefore, we adopted the Swiss-model to construct the structure of P261Q and wild-type TNSALP by means of homology modeling according to your instructions. Alignment of P261Q and wild-type and intramolecular analysis were performed (Page 11, figure 5).

  1. In figure 4, wild-type and mutant models looks very different in part of proteins not affected by the mutation. I suggest that the authors orient the protein in PyMol program and perform the alignment of both wild-type and mutant models to evidence real protein fold changes. In addition, the alignment of both the TNSALP AlphaFold model and the TNSALP homology model (from Swiss_model) is feasible to be done in the Pymol program.

Note: Other programs based on homology modeling could better evidence intra-molecular and inter-molecular interactions, like Swiss-PDBViewer (https://spdbv.unil.ch/). Would be interesting to compare the results of predicted hydrogen bonds and inter-atom distances in both tools.

Note: The conservation and putative location into the domain could bring pieces of evidence to help the authors to support the potential pathogenic status for new mutations. According to Martins et al.. 2019, the P261 is a conserved residue and map in the calcium binding region.

Thank you for the suggestion. It was truly important for us to discuss the potential pathogenicity of p.P261Q. Alignment of P261Q and wild-type TNSALP structures by two methodologies was performed, and no significant changes in secondary and tertiary structures were observed (Page 11, figure 4A). But on further intramolecular analysis new hydrogen bonds between Gln261 and Tyr263 were identified (Page 11, figure 4B, C)

  1. On page 8 - lines 25 to 27. The sentence “...significantly younger at was identified in patients without a history ...” is confusing.

We apologize for the confusion caused. We have rewritten the sentences in the revised version.

(Page 7, line 28-31)

“According to the comparisons between monoallelic heterozygous patients (excluding two compound heterozygous subjects: FAM1-3 and PT-13) with and without a history of fracture (Table 3), the age of patients with fractures was significantly higher than that of patients without fractures (66.8±3.56 vs. 50.75±15.06, p=0.004)”

  1. Is difficult to verify if “...the length of the hydrogen bond was slightly elongated (Figure 4).” (Pag 10, lines 63 and 65). Please, improve the image quality.

We improved the quality of the image and further explored intramolecular contact analysis based on both models constructed by AlphaFold2 (Page 11, figure 4B, C) and Swiss-model (Page 11, figure 5B, C).

Discussion

  1. The authors classify the c.782C>A (p.P261Q) as “unknown” for pathogenic status (table 5), suggesting that no in silico test was performed. However, in the discussion section, the authors report that the clinical significance of c.782C>A (p.P261Q) was still unclear based on 3D models and in silico results. So, the pathogenic status should be inconclusive/inconsistent, but not "unknown". Despite of stated before, they claim that “ ... the pathogeny still exists owing to the proximity of the 261 amino acids to alpha helixes and β-pleated sheets …”. This statement doesn't make sense. What relation is the proximity of the amino acid to alpha helixes and β-sheets and pathogenic potential or pathogenesis?

Thank you for the suggestion. We have replaced “unknown” with “uncertain significance”. In addition, based on your previous suggestions, we have restated and discussed the pathogenicity of mutant proteins in the revised manuscript.

(Page 12-13, line 137-149)

“Therefore, we further established the 3D model of the mutation by AlphaFold 2 and Swiss-model to determine whether the alternation of a single amino acid will lead to structural changes. Significant alterations were not observed in models constructed by both in silico prediction tools. Based on a previous study, proline261 is a highly con-served residue located in the calcium binding region[32]. Therefore, a slight change might lead to significant functional impairment of TNSALP. Proline is classified as a charge-neutral, nonpolar amino acid, while glutamine is a charge-neutral, polar amino acid. This polarity change may alter its interaction with adjacent residues. According to intramolecular contact analysis, new interactions were predicted to arise after the single amino acid alternation (Figure 4B, C), potentially causing impaired function of TNSALP. The clinical manifestations of PT-3 were typical, but the results of bioinformatic analysis were less significant. Therefore, further functional tests at the cellular level are necessary to clarify the pathogenicity of this specific point mutation.”

  1. The statement “Considering the low incidence of de-novo mutations in ALPL, the above differences may be attributed to the founder effect.” (page 13 line 132) doesn't make sense. The authors do not evaluate the geographic origin of mutations.

After careful consideration, we believe that the above discussion is not appropriate. We agree with your valuable suggestions and have deleted the above content.

  1. The statement “They were shorter than their peers.” don’t make sense (page 13 – line 136). Short stature should be defined as a condition in which an individual's height is in the 3rd percentile for the mean height of a given age, sex, and population group.

Thank you for your suggestion. Unfortunately, we currently lack percentile data for the mean height of a given age, sex Chinese population that can be used to compare with the height of our adult HPP patients. In the revised manuscript, we used the average height of age and gender specific populations reported in the 2020 national survey to compare them with our patients.

(Page 13, line 157-159).

“The heights of FAM-1-3 and PT-13 were 132 cm (age-and gender-specific reference av-erage height: 169.7 cm) and 153 cm (age-and gender-specific reference average height: 166.4 cm), respectively”

[1]   Report on Nutrition and Chronic Disease Status of Chinese Residents (2020). People's Medical Publishing House Co., Ltd.: Beijing, 2021.

  1. Abbreviate BTM (page 13 - lines 149 and 159; page 14 – line 213) and AFD (page 14, line 188) were not defined in the article.

    We apologize for the carelessness. The full name of BTMs (bone turnover makers, BTMs) was defined in the Materials and Methods section (Page 2, line 80). “AFD” was a mistype, it should be “AFF”. This was corrected in the revised manuscript (Page 14, line 214).

  1. I consider that the statement “For adult HPP, the severity of the disease has nothing to do with the serum ALP level” (page 13, lines 165-166) is not correct. In the reference cited by the author (Schmidt et al. 2017), the 38 adult patients included in the study received a diagnosis of HPP based on the combination of persistent low serum ALP, raised PLP levels and typical clinical symptoms, and the biochemical analysis performed in this study showed ALP levels below the reference in all cases (Range: 10.0–34.0, Median: 28.5; reference: 35–104 – table 1).

Thank you for your valuable suggestions. We revised our sentences in the manuscript to make our research objective and precise.

(Page 13, line 189-193)

“In adult HPP, the occurrence of fractures and multiple symptoms are associated with increased concentrations of the ALP substrate PLP, while there are no significant differences regarding the level of ALP between patients with and without fracture as well as patients with fewer symptoms and multiple symptoms (more than two symptoms).”

  1. The sentence “the serum level of ALP is always within the normal reference range ...” appear controversial, since those patients were screened based on low serum ALP levels, according to the material and methods section (Page 2, line 60-62).

According to your suggestion, we have modified the relevant content.

(Page 14, line 219-223)

“Both typical clinical manifestations of adult HPP and primary osteoporosis include fragility fracture and low BMD. Therefore, when a patient experiences the following conditions, HPP should be suspected, and further evaluations should be considered to avoid misdiagnosis and underdiagnosis: 1. The serum level of ALP remained consistently lower than 40 U/L

  1. I think that in sentences like “33 reported a debilitating woman with adult-onset HPP, …” (page 14 – line 205) the name of the Author should be explicit in the text and the number of references should appear in the final sentence.

Thank you for the reminder. Author name and a bibliography number were provided in the revised manuscript:

(Page 14, line 229)

“Magdaleno, et al. reported a debilitating woman with adult-onset HPP, whose symptoms of pain and motor function improved significantly after 10 months of treatment with asfotase alfa[46]”

  1. Please, change ALPL for TNSALP or ALP (protein name) in the sentence “Secondly, ALPL substrates,… “ (line 230 page 14).

    We apologize for the mistake. “ALPL” was corrected to “TNSALP”.

  1. In the last paragraph of the discussion, the authors claim that the sample size of the study is limited and not enough to determine the genotype-phenotype correlation of Chinese adult HPP, and in the conclusion, they claim that “This study is the largest sample size study on the clinical and genetic characteristics of HPP in Chinese adults …” (page 14,- line 234-235). This appears controversial.

Thank you for your suggestion. To avoid misunderstandings, we have revised the sentence in the last paragraph of the conclusion:

(Page 15, Line 258-259)

“This study presented and discussed the clinical and genetic characteristics of HPP in Chinese adults and identified three new ALPL mutations.”

Reviewer 2 Report

Clinical and genetic characteristics of hypophosphatasia in Chinese Adults

The manuscript from Xiang Li et al. has provided comprehensive genetic and clinical correlation of patients affected with hypophosphatasia in the 19 Chinese population. They have performed the Sanger sequencing of the ALPL gene and identified the 14 variants (3 novel variants) correlated with the symptoms of the patients. The manuscript is well written. I would advise the author to go through the comments as follows:

Major:

  1. Author should perform classification of the variant based on the guidelines provided by American College of Medical Genetics and Genomics (ACMG-AMP). Based on that, classify the variant as pathogenic, likely pathogenic, Benign and like Benign, and VUS. Author has used a computation prediction tool which is one of the guidelines of the ACMG. Combining all the scores the ACMG tool properly classifies the variant.

  2. The variants are in heterozygous state; it would be interesting to see if their children have inherited the variant and have any effect on them. If the author can identify the variant and they are sure that it is responsible for the disease. The children could be counseled.

  3. The patients that are heterozygous do their parents also suffer with these symptoms as the patients. If yes, it possible to get the DNA from their parents and identify that they have the same variant would help in increasing the chances to be sure about the variant pathogenicity.

  4. Author could show the structural changes using Alphafold2 for each mutation (especially the novel mutation) as they did for P261Q to be more sure that these variants are responsible for disease.

Minor:

  1. In Abstract please mention which gene these mutations belong to. Is it the same gene or different genes that are involved here?

  2. Line 63 ALPL should be in italics. 

  3. Table 3, Female/all sex has only 17 patients. Are we missing 2 patients?

Author Response

Dear Professor,

Thank you for your constructive suggestions. Your valuable advice not only helped us improve the quality of our manuscript but also pointed out the future direction of our subsequent research.

Major

  1. Author should perform classification of the variant based on the guidelines provided by American College of Medical Genetics and Genomics (ACMG-AMP). Based on that, classify the variant as pathogenic, likely pathogenic, Benign and like Benign, and VUS. Author has used a computation prediction tool which is one of the guidelines of the ACMG. Combining all the scores the ACMG tool properly classifies the variant.

Thank you for your valuable suggestions. We reclassified the pathogenicity of variants involved in our study based on the ACMG criteria. The relevant content was updated in Table 5 (Page 8, line 51). The literature that previously reported pathogenic variants according to ACMG guidelines is cited alongside Table 5.

  1. The variants are in heterozygous state; it would be interesting to see if their children have inherited the variant and have any effect on them. If the author can identify the variant and they are sure that it is responsible for the disease. The children could be counseled.

Thank you for your valuable suggestions. Due to the fact that current research is conducted in a retrospective manner, there is a lack of genetic information about patients' immediate families. We agree with your opinion and will collect DNA and biological samples from the patient's relatives during subsequent follow-up studies. The ALPL gene in the patient's family will be sequenced. The levels of ALP substrates - PEA and PLP - will also be measured for all participants. For novel mutations, functional studies will be conducted at the cellular level to determine their pathogenicity. Genetic counseling will be provided to patients and their families.

  1. The patients that are heterozygous do their parents also suffer with these symptoms as the patients. If yes, it possible to get the DNA from their parents and identify that they have the same variant would help in increasing the chances to be sure about the variant pathogenicity.

    We agree with your constructive suggestions. In the next phase of our study, we will investigate both the clinical and genetic characteristics of the patients’ families to consolidate the pathogenicity of each variant.

  1. Author could show the structural changes using Alphafold2 for each mutation (especially the novel mutation) as they did for P261Q to be more sure that these variants are responsible for disease.

Thank you for the constructive suggestion. We predicted the 3-D structures of two remaining novel mutations (p.H171A and p.M467V) by Alphafold2 (Supplementary figure 1). No significant structural changes were found when comparing wild-type TNSALP with the mutant protein (Supplementary figure 2). The results are illustrated in the supporting file (Supporting_file.pdf).

Minor

  1. In Abstract please mention which gene these mutations belong to. Is it the same gene or different genes that are involved here?

Thank you for your suggestion. The only disease-causing gene we discussed in the manuscript was ALPL; therefore we reconstructed our wording and description in the abstract.

(Page 1, Line 15-28)

“Hypophosphatasia (HPP) is an inherited disease caused by ALPL mutation, resulting in decreased alkaline phosphatase (ALP) activity and damage to bone and tooth mineralization. The clinical symptoms of adult HPP are variable, making diagnosis challenging. This study aims to clarify the clinical and genetic characteristics of HPP in Chinese adults. There were 19 patients, including one with childhood-onset and 18 with adult-onset HPP. The median age was 62 (32-74) years, and 16 female patients were involved. Common symptoms included musculoskeletal symptoms (12/19), dental problems (8/19), fractures (7/19), and fatigue (6/19). Nine patients (47.4%) were misdiagnosed with osteoporosis, and six received anti-resorptive treatment. The average serum ALP level was 29.1 (14-53) U/L, and 94.7% (18/19) of patients had ALP levels below 40 U/L. Genetic analysis found 14 ALPL mutations, including three novel mutations - c.511C>G (p.His171Ala), c.782C>A (p.Pro261Gln), and 1399A>G (p.Met467Val). The symptoms of two patients with compound heterozygous mutations were more severe than those with heterozygous mutations. Our study summarized the clinical characteristics of adult HPP patients in the Chinese population, expanded the spectrum of pathogenic mutations, and deepened clinicians' understanding of this neglected disease.”

  1. Line 63 ALPL should be in italics.

    We apologize for the carelessness. We have italicized “ALPL” in Line 63 and rechecked and corrected similar mistakes throughout our manuscript.

  1. Table 3, Female/all sex has only 17 patients. Are we missing 2 patients?

We apologize for the misunderstanding caused. We only compared the laboratory differences of 17 heterozygous (monoallelic) patients with and without a history of fracture in Table 3, because the other 2 compound heterozygous cases (FAM1-3 and PT-13) had significantly severe clinical phenotypes. To avoid confusion, we made a clearer demonstration in the revised manuscript:

(Page 3, line 112-114)

“Student's t test was adopted to compare demographic parameters and laboratory findings between monoallelic heterozygous adult-onset HPP patients with and without a history of fracture.”

(Page 7, line 28-30)

“According to the comparisons between monoallelic heterozygous patients (excluding two compound heterozygous subjects: FAM1-3 and PT-13) with and without a history of fracture (Table 3), the age of patients with fractures was significantly higher than that of patients without fractures (66.8±3.56 vs. 50.75±15.06, p=0.004)”

(Page 7, line 37)

“Table 3. Laboratory differences between monoallelic heterozygous patients with and without fracture.”

Round 2

Reviewer 1 Report

I consider that the authors have substantially improved the manuscript, and it is now suitable for publishing in the Genes periodic.

I have only some small corrections and suggestions.

1) I suggest the authors include the information about previously determinate crystal structure on the homology models based, as well as indicating the PDB number. For example, homology models were based on the crystal structure of human placental alkaline phosphatase (PLAP), PDB number: 1ew2. Note: A correct alignment in the PyMol program requires that both mutant and wild-type should be built based using the same pdb file, for example, 1ew2.

Indicate and/or highlight the mutation position in the TNSALP models (figure 4A and figure 5A) to allow us to evidence structural changes. 

2) Table 2 and table 5 appear before they are mentioned in the text.

3) Standardizing the nomenclature of 25-hydroxy vitamin D in the text. On page 7, line 22, there are two forms, 25OHD and 25(OH)D. 

4) Standardizing the mutation nomenclature in the sentence “c.511C>T (p.H171T) mutation was reported to be pathogenic in LOVD, c.512A>C (p.H171R) was reported to be pathogenic in ClinVar. b c.782C>A (p.Pro261Gln) was previously described by dbSNP (rs765149569) and c.782C>T (p.P261L) was reported to have uncertain significance in ClinVar. c c.1400T>C (p.M467T) was reported to be pathogenic by LOVD.”

Author Response

Dear Professor,

Thank you for your in-depth review and quick response. According to your suggestions, we have revised our manuscript. The point-by-point response is as follows:

1.1 I suggest the authors include the information about previously determinate crystal structure on the homology models based, as well as indicating the PDB number. For example, homology models were based on the crystal structure of human placental alkaline phosphatase (PLAP), PDB number: 1ew2. Note: A correct alignment in the PyMol program requires that both mutant and wild-type should be built based using the same pdb file, for example, 1ew2.

Thank you for the constructive suggestion. We added the above content to the revised version of our manuscript.

(Page 3, line 108-110)

“Homology models were based on the crystal structure of human placental alkaline phosphatase (PDB number: 3mk1.1.A)”

1.2 Indicate and/or highlight the mutation position in the TNSALP models (figure 4A and figure 5A) to allow us to evidence structural changes.

    Thank you for your suggestion. Gln261 was indicated in both Figure 4A (Page 11, line 98) and Figure 5A (Page 12, line 103).

  1. Table 2 and table 5 appear before they are mentioned in the text.

    Thank you for the reminder. We reorganized the structure of our manuscript. Table 2 (Page 7, line 27) and Table 5 (Page 9, line 80) are presented immediately after the text mentioning them.

  1. Standardizing the nomenclature of 25-hydroxy vitamin D in the text. On page 7, line 22, there are two forms, 25OHD and 25(OH)D.

    We apologize for the negligence. We standardized the abbreviation of 25-hydroxy vitamin D as 25OHD.

4) Standardizing the mutation nomenclature in the sentence “c.511C>T (p.H171T) mutation was reported to be pathogenic in LOVD, c.512A>C (p.H171R) was reported to be pathogenic in ClinVar. b c.782C>A (p.Pro261Gln) was previously described by dbSNP (rs765149569) and c.782C>T (p.P261L) was reported to have uncertain significance in ClinVar. c c.1400T>C (p.M467T) was reported to be pathogenic by LOVD.”

Thank you for your careful review. Mutation nomenclature was standardized in the revised version of the manuscript.

(Page 10, line 83-88)

ac.511C>T (p.His171Thr) mutation was reported to be pathogenic in ¬LOVD, c.512A>C (p.His171Arg) was reported to be pathogenic in ClinVar. Evidence of ACMG classification: PM2 + PM5 + PP3 + PP2 + PP4. bc.782C>A (p.Pro261Gln) was previously described by dbSNP (rs765149569) and c.782C>T (p.Pro261Leu) was reported to have uncertain significance in ClinVar. Evidence of ACMG classification: BP6 + PP2 + PP4. cc.1400T>C (p.Met467Thr) was reported to be pathogenic by LOVD. Evidence of ACMG classification: PM2 + PM5 + PP3 + PP2 +

Reviewer 2 Report

The authors has changed the manuscript according to the comments.

The changes authors mentioned they put it in the supplementary files which I could not find. Please upload it.

For ACMG, I understand that the authors has found the variants but these variants are not classified based on the ACMG guidelines in the literature. Please do it.

Author Response

Dear Professor,

Thank you for your in-depth review and quick response. According to your suggestions, we have revised our manuscript. The point-by-point response is as follows:

  1. The changes authors mentioned they put it in the supplementary files which I could not find. Please upload it.

We apologize for the inconvenience we brought. Actually, we have uploaded the supporting file with our response in round 1, as “author-coverletter-28004871.v1.pdf” (the file name was automatically generated). Meanwhile, you can download the re-uploaded file by clicking “author-coverletter-28598462.v1.pdf” in the Author's Reply to the Review Report section or “Report Notes” in the Authors' Responses to Reviewer's Comments section since we find it impossible to insert those images in the webpage.

  1. For ACMG, I understand that the authors has found the variants but these variants are not classified based on the ACMG guidelines in the literature. Please do it.

We apologize for the ambiguous statement we made in the manuscript. Under your guidance, evidence of ACMG classification was provided with each novel mutation we identified in the current study.

(Page 10, line 83-88)

ac.511C>T (p.His171Thr) mutation was reported to be pathogenic in ¬LOVD, c.512A>C (p.His171Arg) was reported to be pathogenic in ClinVar. Evidence of ACMG classification: PM2 + PM5 + PP3 + PP2 + PP4. bc.782C>A (p.Pro261Gln) was previously described by dbSNP (rs765149569) and c.782C>T (p.Pro261Leu) was reported to have uncertain significance in ClinVar. Evidence of ACMG classification: BP6 + PP2 + PP4. cc.1400T>C (p.Met467Thr) was reported to be pathogenic by LOVD. Evidence of ACMG classification: PM2 + PM5 + PP3 + PP2 + PP4.
